# Chemical Compositions of Lianqiao (*Forsythia suspensa*) Extracts and Their Potential Health Benefits

**DOI:** 10.3390/ph17060740

**Published:** 2024-06-06

**Authors:** Boyan Gao, Hanshu Zhu, Zhihao Liu, Xiaohua He, Jianghao Sun, Yanfang Li, Xianli Wu, Pamela Pehrsson, Yaqiong Zhang, Yuanhang Yao, Liangli Yu

**Affiliations:** 1Institute of Food and Nutraceutical Science, School of Agriculture and Biology, Shanghai Jiao Tong University, Shanghai 200240, China; gaoboyan@sjtu.edu.cn (B.G.); zhuhanshu@sjtu.edu.cn (H.Z.); 2Department of Nutrition and Food Science, University of Maryland, College Park, MD 20742, USA; zhihao.liu@usda.gov (Z.L.); yfl0820@umd.edu (Y.L.); lyu5@umd.edu (L.Y.); 3Methods and Application of Food Composition Laboratory, Beltsville Human Nutrition Research Center, Agricultural Research Service, United States Department of Agriculture, Beltsville, MD 20705, USA; jianghao.sun@usda.gov (J.S.); xianli.wu@usda.gov (X.W.); pamela.pehrsson@usda.gov (P.P.); 4Western Regional Research Center, Agricultural Research Service, United States Department of Agriculture, Albany, CA 94710, USA; xiaohua.he@usda.gov

**Keywords:** HPLC-MS/MS, radical scavenging, COVID-19, SARS-CoV-2 virus spike protein, ACE2, *Forsythia suspensa* (Lianqiao)

## Abstract

This study evaluated the fruits of *Forsythia suspensa* (Lianqiao), an important economic crop, for the chemical components of its water and ethanol extracts, inhibitory effects on SARS-CoV-2 virus spike protein binding to ACE2, inhibition of ACE2 activity, and capacity to scavenge free radicals. A total of 42 compounds were tentatively identified in the extracts via HPLC-MS/MS analysis. The water extract showed a greater ACE2 inhibition but a weaker inhibition on SARS-CoV-2 spike protein binding to ACE2 than the ethanol extract on a per-botanical-weight-concentration basis. The phenolic content was found to be greater in the water extract at 45.19 mg GAE/g dry botanical weight than in the ethanol extract (6.89 mg GAE/g dry botanical). Furthermore, the water extract had greater scavenging capacities against HO^●^, DPPH^●^, and ABTS^●+^ at 448.48, 66.36, and 121.29 µmol TE/g dry botanical, respectively, as compared to that of the ethanol extract (154.04, 3.55, and 33.83 µmol TE/g dry botanical, respectively). These results warrant further research into, and the development of, the potential COVID-19-preventive applications of Lianqiao and its extracts.

## 1. Introduction

According to the World Health Organization (WHO), severe acute respiratory syndrome coronavirus (SARS-CoV-2), responsible for the outbreak of the coronavirus disease 2019 (COVID-19) pandemic, has infected 762,791,152 people and led to 6,897,025 deaths globally [1]. Beyond the vaccines and pharmaceuticals, there is an emerging research trend that has started to direct attention to some alternative remedies, such as utilizing certain bioactive compounds, nutraceuticals, and herbs in reducing the risk of infection and/or relieving the symptoms of SARS-CoV-2. *Forsythia suspensa*, an important agribotanical primarily grown in China, is well known for its fruit, Lianqiao (*Forsythiae fructus*), which has been an essential component of antibacterial remedies for decades. It contains many bioactive compounds, including lignans, phenylethanoid glycosides, flavonoids, terpenoids, cyclohexylethanol derivatives, alkaloids, and steroids [2], which endow Lianqiao with antioxidant [3], antibacterial [4], and antiviral [5] activities.

Law et al. reported that Lianqiao extracts prolonged the life span and increased the survival of mice infected with the A/PR/8/34 (H1N1) virus in a dose-dependent manner [6]. Lin et al. collected the clinical data of 77 patients with severe acute respiratory syndrome (SARS) treated with integrated regular and alternative medicines, including botanical foods, and the results showed that the administration of botanical foods, including Lianqiao, had potential in preventing severe SARS cases [7]. Considering the similarity between severe acute respiratory syndrome coronavirus (SARS-CoV-2) spike protein and SARS-CoV spike protein (around 76% to 78%), in ref. [8], the antiviral and anti-inflammatory activity of Lianqiao-related medicine were examined, and results showed that Lianqiao has significant inhibitory effects against the SARS-CoV-2 replication and shows anti-inflammatory activity in vitro [9], which possibly implies that Lianqiao could have potential effects in reducing SARS-CoV-2 infection and the progression of severe COVID-19 symptoms.

According to its pathogenic mechanism, SARS-CoV-2 enters host cells in the same way as SARS-CoV via the angiotensin-converting enzyme 2 (ACE2) receptor. After the binding of SARS-CoV-2 spike (S) protein to the ACE2, the membrane of the virus and the host cell fuses, and the viral RNA is subsequently released into the cytoplasm of the host cell to establish infection [10]. The sites involved in this process might be focused as targets for the development of COVID-19 prevention and/or therapeutics. For example, recombinant human (rh) ACE2 is an ACE2 blocker binding to virus S protein, thus protecting host cells from infection [11].

Although some preliminary studies have suggested that Lianqiao might be one of the possible candidates for reducing the risk of COVID-19 and the severity of its symptoms [9,12], its action site and direct antiviral effects have never been investigated. Therefore, this study was conducted to investigate the chemical compositions of Lianqiao water and ethanol extracts, their potential effects on the binding of SARS-CoV-2 spike protein to ACE2, their inhibition of ACE2 activity, and their scavenging capacities against hydroxyl (HO^●^), DPPH^●^, and ABTS^●+^ radicals. The present study aims to elucidate the inhibitory effects of Lianqiao on ACE2-mediated viral invasion and can hopefully provide scientific support for the development of Lianqiao and its derivatives as functional foods for COVID-19 prevention.

## 2. Results and Discussion

Chemical compositions of the Lianqiao (*Forsythia suspensa* fruit) extracts: The LC-MS chromatograms of Lianqiao water and ethanol extracts in positive- and negative-ionization modes are shown in Appendix A. A total of 42 compounds were tentatively identified referring to high-resolution MS, MS^2^ fragment information, and a literature review. Among these compounds, there were 8 lignans (**30**, **32**–**35**, **37**, **38**, **40**), 9 phenylethanoid glycosides (**4**, **9**, **13**, **18**, **20**, **26**–**29**), 5 terpenoids (**16**, **36**, **39**, **41**, **42**), 11 phenolics (**1**, **3**, **5**, **8**, **11**, **12**, **14**, **19**, **21**, **23**, **25**), and 9 other compounds (Table 1). As shown in Table 1, Lianqiao samples with different extraction solvents were observed with varied identified compounds and different relative contents. Generally, 37 compounds were detected and identified in Lianqiao water extracts, and 25 compounds were identified in ethanol extracts. Some identified compounds had the same elemental composition, molecular formula, and similar fragment information, making it impossible to determine their specific substance attribution, and they were all marked as isomers in this article. For the relative contents, halleridone (**7**), forsythialan B (**35**), (+)-1-hydroxylpinoresinol (**33**), plantasioside (**24**), and forsythoside H (**26**) were the five major compounds in water extracts; whereas forsythialan B (**35**), (+)-1-hydroxylpinoresinol (**33**), azelaic acid (**31**), epipinoresinol-4’-β-D-glucopyranoside (**32**), and forsythoside H (**26**) were the top five ethanol extracts. Forsythialan B (**35**), (+)-1-hydroxylpinoresinol (**33**), and forsythoside H (**26**) could be recognized as the three primary compounds in both Lianqiao water and ethanol extracts. Compounds **28**, **35**, and **18** were selected as examples with which to elucidate the identification progress. The rest of the major chemical components in the Lianqiao extracts were identified consistently using the approaches described below.

For compound **28**, the molecular ion detected in the positive-ion mode ([M + H]^+^) is *m*/*z* 625.21246, which is consistent with C_29_H_37_O_15_ (mass precision, −0.379 ppm), indicating that the molecular formula of compound **28** is C_29_H_36_O_15_. The main fragment ions in the normal mode are 479.1540 [M + H − Ara]^+^, 471.1494 [M + H − C_8_H_10_O_3_]^+^, and 325.0915 [M + H − Ara − C_8_H_10_O_3_]^+^. According to these results and the literature, compound **28** was identified as forsythoside A, a type of phenylethanoid glycosides. Forsythoside A has been proven to have antioxidant, anti-inflammatory, and antiviral activities, and is known as one of the representative active substances in Lianqiao [17,18,19]. The content of forsythoside has been designated as a marker for Lianqiao’s quality control according to *Chinese Pharmacopoeia* (2015 Edition) [20].

For compound **35**, in the positive-ion mode ([M + H]^+^), the molecular ion is *m*/*z* 389.15921, which is consistent with C_21_H_25_O_7_ (mass accuracy: −0.693 ppm). And in the negative-ion mode ([M − H]^–^) the molecular ion is *m*/*z* 387.14408, which is consistent with C_21_H_23_O_7_ (mass accuracy: 0.647 ppm). These results indicate that the molecular formula of compound **35** is C_21_H_24_O_7_. The fragment ions detected in the positive mode are at *m*/*z* 371.1485 [M + H − H_2_O]^+^, 247.0963 [M + H − H_2_O − 124]^+^, and 217.0858 [M + H − H_2_O − 124 − CH_2_O]^+^. The fragment ions of [M + H − H_2_O − 124]^+^ are formed by the reduction of methoxyphenol group. The major fragment ion in the negative mode is 372.1202 [M − H − CH_3_]^–^. Based on these results and the literature review, compound **35** was identified as forsythialan B. Forsythialan B is a tetrahydrofuran, one of the representative lignans in Lianqiao, which has been proven to be protective against peroxynitrite-induced oxidative stress [21].

For compound **18** (Figure 1), the molecular ion is *m*/*z* 487.14523 in the negative mode ([M − H]^–^), which is consistent with C_21_H_27_O_13_ (mass accuracy: 1.258 ppm) and indicates its molecular formula of C_21_H_28_O_13_. The fragment ions detected in the negative-ion mode are at *m*/*z* 469.1335 [M − H − H_2_O]^–^, 427.1229 [M − H − H_2_O − C_2_H_2_O]^–^, 397.1122 [M − H − H_2_O − C_2_H_2_O − CH_2_O]^−^, 233.0445 [M − H − H_2_O − C_2_H_2_O − CH_2_O − O-Ara]^–^, 203.0340 [M − H − H_2_O − C_2_H_2_O − CH_2_O − O-Ara − CH_2_O]^−^, and 179.0341 [M − H − H_2_O − C_2_H_2_O − CH_2_O − O-Ara − CH_2_O − C_2_]^−^. Based on the data we collected and the existing evidence from the literature, compound **18** was tentatively identified as cistanoside F. To the best of our knowledge, cistanoside F was found in Lianqiao extracts for the first time, but it has been found in other botanicals before [22,23].

Inhibitory effects of Lianqiao extracts on SARS-CoV-2 spike protein and ACE2 interaction: Significant differences were observed in the inhibitory effects of SARS-CoV-2 spike protein and ACE2 interaction between Lianqiao water and ethanol extracts (Figure 2). Notably, the water extract at 33.3 mg dry botanical equivalents/mL in the testing mixture (WE33.3) inhibited 84.34% of the interaction activity. Meanwhile, 3.3 mg dry botanical equivalents/mL in the testing mixture (WE3.3) inhibited only 1.43% of the interaction activity, while the ethanol extracts (EE3.3) inhibited 45.57% of the interaction activity at the same concentration. At a concentration of 1.7 mg dried botanical equivalent/mL, the ethanol extracts (EE1.7) were observed with a 7.45% inhibition of SARS-CoV-2 spike protein binding to ACE2. However, the inhibitory effects were negligible, with a similar concentration of the water extracts. This is possible because the ethanol extracts of Lianqiao exhibit greater activity in inactivating and denaturing the SARS-CoV-2 virus [24]. Given that SARS-CoV-2 enters host cells by binding ACE2 as the receptor, intervention and inhibition of this process may be one of the key points for preventing infection and relieving the symptoms of COVID-19.

Inhibitory effects of the Lianqiao extracts on ACE2 enzyme activity: Both Lianqiao water and ethanol extracts exhibited inhibitory effects on ACE2 activity (Figure 3). Specifically, the initial water extracts (5.0 mg dry botanical equivalents/mL in the testing mixture, WE5.0) inhibited 77.82% of ACE2 activity, and an inhibition of 14.93% was observed with the initial water extracts at a concentration of 0.5 mg dry botanical equivalents/mL (WE0.5). In contrast, the initial ethanol extracts (5.0 mg dry botanical equivalents/mL, EE5.0) inhibited 68.33% of ACE2 activity, which was about 12.2% lower than the water extracts at the same concentration. Current evidence suggests that ACE2 is the receptor by which SARS-CoV-2 enters host cells, and coronavirus aggravates infection by upregulating ACE2 expression via interferon [25,26]. Therefore, the inhibition of its own activity may be one approach to interfering with the infection process. However, it is worth noting that ACE2 is also a critical component of the renin–angiotensin system (RAS), which regulates pathological responses such as blood pressure, inflammation, and oxidative stress [10,27]. And further research is still needed to determine the possible inhibition effects of ACE2 activity via Lianqiao extracts in simultaneously inducing some other side effects.

Total phenolic contents (TPC) and antioxidant assays of Lianqiao extracts: Phenolics are capable of scavenging free radicals or chelating metals, which competitively inhibit the oxidation process [28]. The TPC values of Lianqiao water and ethanol extracts were 45.19 and 6.89 mg GAE/g dry botanical, respectively (Figure 4). The TPC values of the Lianqiao water extracts in the present study were more remarkable than those of a previous study, which were reported to be 13.2–13.9 mg GAE/g dry botanical from Lianqiao methanol (70%, *v*/*v*) extracts [3]. These results indicate that different extraction methods might lead to substantial differences in the TPC of Lianqiao extracts, which aligns with our chemical profile findings.

Free radicals in biological systems can damage a variety of cell components and cause adverse effects. For instance, peroxidation of membrane phospholipids changes the fluidity and permeability of cell membranes, leading to cell dysfunction or even death, which will thereby result in metabolic disorders and a variety of diseases [29]. Meanwhile, the reactive oxygen species (ROS) have also been discovered to be involved in the pathogenesis of neurodegenerative diseases [30], atherosclerosis [31], asthma [32], and chronic kidney diseases [33]. In COVID-19, SARS-CoV-2 unbalances oxidative homeostasis and stimulates the production of ROS, triggers an inflammatory reaction, and then causes organ failure [34]. Cytokine storm, a severe inflammatory reaction mediated by oxidative stress, leads to even more severe symptoms [25,34]. Therefore, evaluating the antioxidant abilities of certain components is essential when developing potential nutraceuticals for COVID-19 treatment. Considering the difference in mechanisms of action and the targeted free radicals, three free-radical-scavenging assays, including HOSC, RDSC, and ABTS, were selected to examine the in vitro antioxidant effects of Lianqiao extracts.

The free-radical-scavenging capacities of Lianqiao water and ethanol extracts are shown in Figure 5. The HOSC, RDSC, and ABTS values of the water extracts were 448.48, 66.36, and 121.29 µmol TE/g, respectively, which were significantly greater than those of the ethanol extracts (154.04, 3.55, and 33.83 µmol TE/g, respectively). Overall, the results suggested that both Lianqiao water and ethanol extracts possessed considerable antioxidant capacities, whereas Lianqiao water extracts had a superior antioxidant capacity than ethanol extract. The higher antioxidant capacity of Lianqiao water extract might be due to its higher total phenolic content shown in Figure 4. Specifically, for the HOSC assay, the extracts of Lianqiao showed significant efficacy in scavenging hydroxyl radicals, which is of clinical importance due to the critical role that hydroxyl radicals play in irreversible damage caused by oxidative stress in our body [35]. It is worth noting that the antiradical activity inhibited by using stable free radicals, such as DPPH and ABTS, may not be fully representative of the antioxidant activity of the tested samples, and they do not exist in biological systems [35]. Therefore, further research is needed to validate the antioxidant activity of Lianqiao extracts with an in vivo study design.

## 3. Materials and Methods

Materials: Lianqiao samples (*Forsythia suspensa* fruit) of Chinese Pharmacopeia grade were obtained from a local pharmacy (Rockville, MD, USA). Antioxidant-assays-related chemical reagents, including the Folin–Ciocalteu reagent (FC reagent), sodium carbonate (Na_2_CO_3_), gallic acid, 6-hydroxy-2,5,7,8-tetramethylchroman-2-carboxylic acid (Trolox), fluorescein (FL), disodium hydrogen phosphate dodecahydrate (Na_2_HPO_4_·12H_2_O), sodium dihydrogen phosphate dehydrate (NaH_2_PO_4_), ferric chloride (FeCl_3_), hydrogen peroxide (H_2_O_2_) (30%), 2,2-diphenyl-1-picrylhydrazyl (DPPH^•^), and 2,2′azinobis (3-ethylbenzothiazoline-6-sulfonic acid) diammonium salt (ABTS), were purchased from Sigma-Aldrich (St. Louis, MO, USA). Formic acid and acetonitrile (LC-MS grade) were obtained from Merck (Darmstadt, Germany). ACE2 Inhibitor Screening Assay Kit (No. 502100) and SARS-CoV-2 spike ACE2 interaction Inhibitor Screening Assay Kit (No. 502050) were purchased from Cayman (Ann Arbor, MI, USA). All other chemicals used in this study were of analytical-grade purity and purchased from Fisher Scientific (Hampton, NH, USA) without further processing.

Sample preparation and extraction: Lianqiao fruit samples were ground using a micromill grinder (Bel Art Products, Pequannock, NJ, USA) to a particle size of 0.42 mm powder. For water extraction, Lianqiao powder was extracted with ultrapure water in a water bath at 85 °C for an initial 2 h (1:10, *w*/*v*) and was then extracted for 22 h at ambient temperature. For ethanol extraction, the powder was extracted with pure ethanol for an entire 24 h at ambient temperature (1:10, *w*/*v*). Then, water and ethanol extracts were centrifuged, and the supernatants were stored at −20 °C before analyses.

Chemical compositions of Lianqiao (*Forsythia suspensa* fruit): To separate and analyze the chemical compositions of Lianqiao water and ethanol extracts, a Vanquish UHPLC (Thermo Fisher Scientific, Norristown, PA, USA) and an Orbitrap Fusion ID-X Tribrid mass spectrometer (Thermo Fisher Scientific, Norristown, PA, USA) were used. Chemical components in Lianqiao extract samples were separated using an Agilent Eclipse Plus-C_18_ UHPLC column (150 mm × 2.1 mm, 1.8 μm) (Santa Clara, CA, USA) connected after an UltraShield pre-column (Santa Clara, CA, USA), with an injection volume of 1 μL. And 0.1% formic acid in water (*v*/*v*) and 0.1% formic acid in acetonitrile (*v*/*v*) were used as mobile phases A and B, respectively. The column was pre-equilibrated with 2% B for 5 min; then the linear gradient was programmed as follows: 0 min, 2% B; 15 min, 10% B; 35 min, 40% B; and 55 min, 95% B, lasting until 60 min and then re-equilibrated the column with 2% B for 10 min. The flow rate was 0.3 mL/min throughout the injection. For the high-resolution mass spectrometry analyses, the mass scanning process took 39 min in the scan range of *m*/*z* 120−1200. The spray voltages in positive- and negative-ion modes were set to 3900 and 2500 V, respectively. The temperature of the ion transport tube was kept at 300 °C, and the temperature of the vaporizer was 275 °C. The chemical compounds in Lianqiao water and ethanol extracts were determined by their molecular weights, mass fragment information, previously published related manuscripts, and chemical databases, including SciFinder and ChemSpider.

Inhibitory effects of Lianqiao extracts on SARS-CoV-2 spike protein and ACE2 interaction: To detect the inhibitory effects of Lianqiao extracts on the binding of SARS-CoV-2 spike protein and ACE2, a Cayman SARS-CoV-2 spike ACE2 interaction inhibitor screening assay kit was used (No. 502050). The procedure was performed following the protocol of the product. Briefly, 100 µL of buffer (Immunoassay Buffer C (1X)) and 50 µL of ACE2 inhibitor screening reagent were added to a 96-well plate as the background. For the control sample, 50 µL of buffer, 50 µL of ACE2 inhibitor screening reagent, and 50 µL of spike inhibitor screening reagent were added to the 100% initial activity wells. For the tested sample, 50 µL of Lianqiao extract sample solution, 50 µL ACE2 inhibitor screening reagent, and 50 µL of spike inhibitor screening reagent were added to the sample wells. The 96-well plate was covered and incubated for 60 min at ambient temperature with shaking. Then, the plate was emptied and rinsed five times with wash buffer; then, 150 µL of anti-His HRP conjugate was added. The plate was incubated at room temperature with shaking for 30 min, and the washing procedure was repeated. A volume of 175 µL of TMB substrate solution was added to each well, and the plate was incubated again at ambient temperature with shaking for 15 min. A volume of 75 µL of HRP stop solution was added to each well to terminate the reaction, and absorbance was read at 450 nm using Tecan M200 Pro microplate reader (Tecan Group Ltd., Mannedorf, Switzerland). The results were shown as the percentage (%) inhibition:$$\%inhibition=\frac{{Abs}_{Sample}-{Abs}_{Background}}{{Abs}_{100\%Initial}-{Abs}_{Background}}\times 100\%.$$

Inhibitory effects of Lianqiao extracts on ACE2 enzyme activity: A Cayman ACE2 Inhibitor Screening Assay Kit (No. 502100) was used to determine the ACE2 inhibitory effects of Lianqiao extracts. The procedure was performed following the protocol of the product. Briefly, 5 µL solvent and 85 µL ACE2 assay buffer were added to the background wells. For the control reaction, 5 µL solvent, 75 µL ACE2 assay buffer, and 10 µL of ACE2 enzyme were added to the 100% initial activity wells. And 5 µL of Lianqiao extract sample solution, 75 µL of ACE2 assay buffer, and 10 µL diluted ACE2 enzyme were added to the sample reaction wells. A volume of 10 µL of ACE2 substrate was added to all wells to start the reactions. The 96-well plate was covered and incubated for 30 min at ambient temperature in the dark. After the incubation, the fluorescence (AF) (λ_ex_ = 320 nm; λ_em_ = 405 nm) of the plate was read using a Tecan M200 Pro microplate reader (Tecan Group Ltd., Mannedorf, Switzerland). The results were shown as the percentage (%) inhibition:$$\%inhibition=\left(1-\frac{{AF}_{Sample}-{AF}_{Background}}{{AF}_{100\%Initial}-{AF}_{Background}}\right)\times 100\%.$$

Total phenolic content (TPC) determination: Total phenolic contents (TPC) of Lianqiao water and ethanol extracts were examined following the laboratory protocol [36]. One aliquot of 3 mL of ultrapure water, 50 µL of Lianqiao extraction sample, gallic acid standards or solvent, and 250 µL of FC reagent were added to a test tube and vortexed for 5 s to mix well. After waiting 5 min, 750 µL of 20% (*w*/*v*) Na_2_CO_3_ was added to activate the reaction. The tubes were covered with parafilm and aluminum foil for 2 h at ambient temperature in the dark. Then, the spectrophotometer was blanked at 765 nm using a blank solution, and the absorbances of all standard and sample solutions at 765 nm were measured with a multifunction microplate reader (Tecan M200 Pro, Tecan Group Ltd., Mannedorf, Switzerland). The TPC of Lianqiao water and ethanol extracts was showed as milligram gallic acid equivalents per gram of Lianqiao sample (mg GAE/g).

Relative hydroxy radical scavenging capacity (HOSC): HOSC values of the Lianqiao water and ethanol extracts were tested following the laboratory protocol [37]. Fresh prepared working solutions, including 170 µL of working FL solution, 30 µL of sample, Trolox standards or blank, 40 µL of H_2_O_2_ working solution, and 60 µL of FeCl_3_ working solution were mixed in a 96-well plate. After shaking for 15 s, the fluorescence intensities (λ_ex_ = 485 nm; λ_em_ = 528 nm) of each well were measured every 5 min for 5 h by Tecan M200 Pro microplate reader (Tecan Group Ltd., Mannedorf, Switzerland). The area under the curve (AUC) was measured, and the HOSC values of Lianqiao extracts were showed as Trolox equivalent per gram of Lianqiao sample (µmoles TE/g).

Relative DPPH^●^ scavenging capacity (RDSC): The RDSC values of the Lianqiao water and ethanol extracts were determined following our previously published protocol [38]. In a 96-well plate, 200 µL of solvent was used as blank, 100 µL of the sample, and Trolox standards or solvent were added together with 100 µL of DPPH^●^ working solution (0.2 mM). The absorbance at 515 nm was measured every minute for 90 min using Tecan M200 Pro microplate reader (Tecan Group Ltd., Mannedorf, Switzerland). After calculating the AUCs, the RDSC values of Lianqiao extracts were expressed as µmol Trolox equivalent per gram of Lianqiao sample (µmoles TE/g).

Relative ABTS^●+^ scavenging capacity (ABTS): The ABTS values of the Lianqiao water and ethanol extracts were tested [39]. The ABTS^•+^ working solution was diluted to absorbance at 0.700 with the monitor of spectrometer at 734 nm. Then 2 mL ABTS^•+^ working solution, 160 µL of sample, or Trolox standards were mixed to start the reaction and vortexed for 30 s. The mixture was transferred to a cuvette, and the absorbance at 734 nm was measured exactly 90 s after the initial of reaction with a Genesys 20 visible spectrophotometer (Thermo Fisher Scientific, Norristown, PA, USA). The ABTS values of Lianqiao water and ethanol extracts were showed as µmol Trolox equivalent per gram of Lianqiao sample (µmoles TE/g).

Statistical analysis: All the experiments were carried out in triplicate, and the data were presented as mean ± standard deviation (SD). The *t*-test was conducted to determine the differences in the inhibitory and radical scavenging activities and the total phenolic content between Lianqiao water and ethanol extracts using IBM SPSS Statistics (version 25.0, SPSS, Inc., Chicago, IL, USA), and *p* < 0.05 was considered statistically significant. The mass data and spectra were obtained and analyzed by Xcalibur^TM^ (Version 4.2, Thermo Fisher Scientific, Norristown, PA, USA). All the figures were charted using GraphPad Prism (version 8.0, Graphpad Software Inc., San Diego, CA, USA). 

## 4. Conclusions

In conclusion, this present study managed to identify the chemical components of Lianqiao water and ethanol extracts, and Cistanoside F was found in Lianqiao extracts for the first time. This study has contributed to a preliminarily understanding of the effects of Lianqiao water and ethanol extracts in preventing and/or treating SARS-CoV-2-related infections by evaluating the SARS-CoV-2 spike protein and ACE2 interaction, ACE2 enzyme activity, and antioxidant capacities. The Lianqiao water extract showed stronger inhibition on the activity of ACE2 than the ethanol extract, but its inhibitory effects on the binding of SARS-CoV-2 spike protein and ACE2 was weaker than that of ethanol. In addition, the scavenging capacities of water extract on HO^●^, DPPH^●^, and ABTS^●+^ were stronger than those of ethanol extract. These suggest that the extraction solvents matter and are likely to result in different extracted components and bioactivities. Taken together, this study provides scientific support for the application of Lianqiao as a potential integrant for the prevention and treatment of SARS-CoV-2 infection and contributes to the study of an alternative remedy for COVID-19 utilizing Lianqiao extracts. 

## Figures and Tables

**Figure 1 pharmaceuticals-17-00740-f001:**
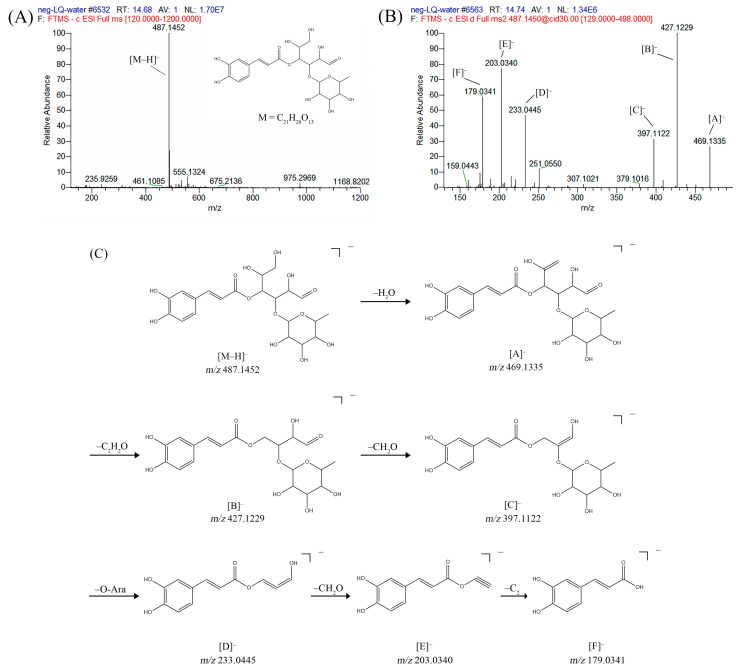
Identification of cistanoside F (compound **18**). Full-scan MS (**A**) and MS^2^ (**B**) in negative-ionization mode—summarization of inferred fragmentation approaches of [M − H]^−^ at *m*/*z* 487.1452 (**C**).

**Figure 2 pharmaceuticals-17-00740-f002:**
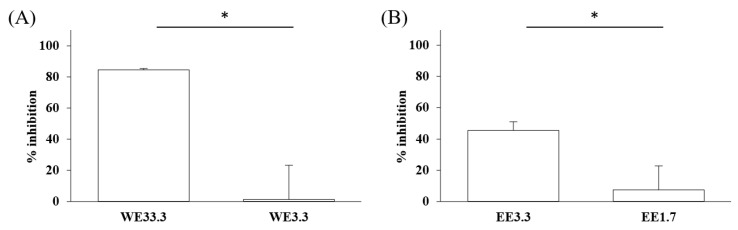
Inhibitory effects of Lianqiao (**A**) water and (**B**) ethanol extracts at different concentrations on binding interaction between SARS-CoV-2 spike protein and ACE2. WE33.3 and WE3.3 stand for the water extracts at concentrations of 33.3 and 3.3 mg dry Lianqiao equivalents/mL in the testing mixture, respectively. EE3.3 and EE1.7 stand for ethanol extracts at concentrations of 3.3 and 1.7 mg dry Lianqiao equivalents/mL in the testing mixture, respectively. Results are expressed as mean ± SD of experiments performed in triplicate, and values are on a dry botanical weight basis. * indicates significant differences (*p* < 0.05).

**Figure 3 pharmaceuticals-17-00740-f003:**
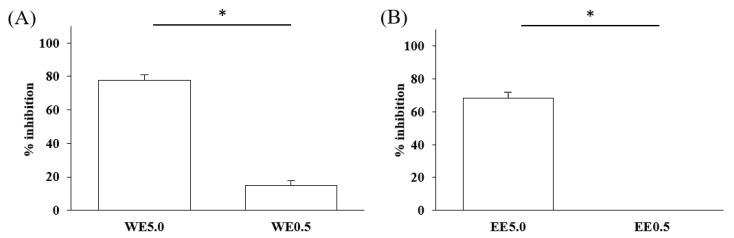
ACE2 inhibitory effects of Lianqiao (**A**) water and (**B**) ethanol extracts at different concentrations. WE5.0 and WE0.5 stand for the water extracts at concentrations of 5.0 and 0.5 mg dry Lianqiao equivalents/mL in the testing mixture, respectively; EE5.0 and EE0.5 stand for the ethanol extracts at concentrations of 5.0 and 0.5 mg dry Lianqiao equivalents/mL in the testing mixture, respectively. Results are expressed as mean ± SD of experiments performed in triplicate, and values are on a dry botanical weight basis. * indicates significant differences (*p* < 0.05).

**Figure 4 pharmaceuticals-17-00740-f004:**
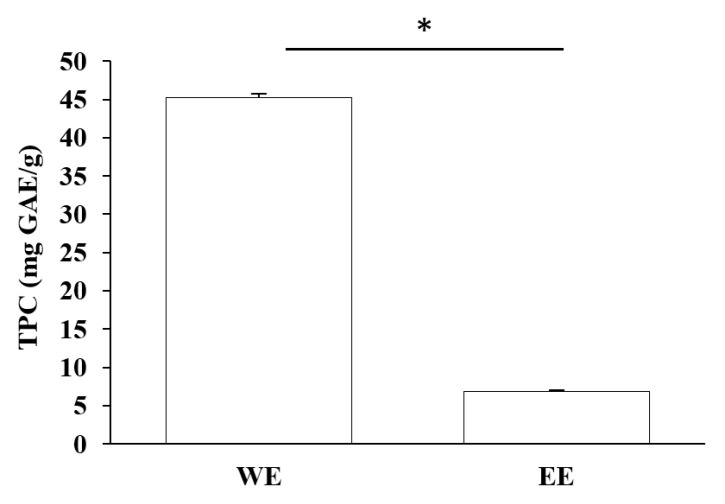
Total phenolic content (TPC) of Lianqiao water (WE) and ethanol extracts (EE). Results are expressed as mean ± SD of experiments performed in triplicate, and values are on a dry botanical weight basis. * indicates significant differences (*p* < 0.05).

**Figure 5 pharmaceuticals-17-00740-f005:**
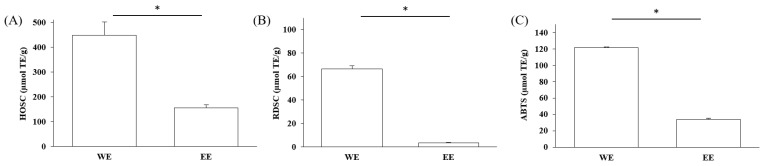
Relative free radical scavenging activities of Lianqiao water (WE) and ethanol extracts (EE) against (**A**) HO^●^, (**B**) DPPH^●^, and (**C**) ABTS^●+^. Results are expressed as mean ± SD of experiments performed in triplicate, and values are on a dry botanical weight basis. * indicates significant differences (*p* < 0.05).

**Table 1 pharmaceuticals-17-00740-t001:** Characterization of compounds identified in *Forsythia suspensa* fruit (Lianqiao).

ID	Positive Mode (ESI^+^)	Negative Mode (ESI^−^)	Formula	Name	Structure	Relative Ion Intensity (×10^7^)	Ref.
Retention Time	Exptl.[M + H]^+^	Fragment Ions	Mass Error (ppm)	Retention Time	Exptl.[M − H]^−^	Fragment Ions	Mass Error (ppm)	WE(+/−)	EE(+/−)
1.	1.03	127.03889	nd	−0.635	nd	nd	nd	nd	C_6_H_6_O_3_	benzene-1,2,4-triol	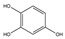	1.03/nd	nd/nd	[13]
2.	2.78	268.10385	136.0612	−0.673	nd	nd	nd	nd	C_10_H_13_O_4_N_5_	adenosine	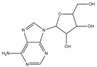	5.47/nd	nd/nd	[14]
3.	3.14	169.04961	151.0385,123.0436	0.442	3.24	167.03466	123.0447,108.0213	4.638	C_8_H_8_O_4_	vanillic acid	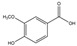	1.06/7.54	2.34/nd	[14]
4.	3.55	317.12292	299.1143,263.0929,221.0820,**137.0603**	−0.549	3.72	315.10797	**179.0567**,161.0459	1.669	C_14_H_20_O_8_	isomer of hydroxytyrosol glucoside	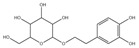	4.58/2.46	nd/nd	[14]
5.	3.82	165.05450	**147.0434**,119.0486	−0.731	nd	nd	nd	nd	C_9_H_8_O_3_	isomer of p-Coumaric acid	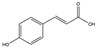	7.52/nd	nd/nd	
6.	nd	nd	nd	nd	4.34	191.05573	145.0500,129.0552,**101.0602**	3.745	C_7_H_12_O_6_	quinic acid	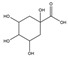	nd/21.27	nd/nd	[13]
7.	4.83	155.07012	137.0592,109.0644	−0.972	nd	nd	nd	nd	C_8_H_10_O_3_	halleridone	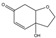	38.28/nd	nd/nd	[13]
8.	nd	nd	nd	nd	5.74	153.01794	109.0284	−1.929	C_7_H_6_O_4_	protocatechuic acid	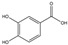	nd/nd	nd/0.50	[15]
9.	6.49	317.12283	**299.1146**,281.1040	−0.833	nd	nd	nd	nd	C_14_H_20_O_8_	isomer of hydroxytyrosol glucoside	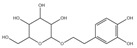	1.00/nd	nd/nd	[2]
10.	nd	nd	nd	nd	7.89	315.07126	**153.0197**,109.0295	0.640	C_13_H_16_O_9_	protocatechuic acid-O-glucopyranoside	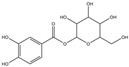	nd/0.76	nd/nd	[14]
11.	9.40	139.03885	111.0436,93.0694	−0.867	10.15	137.02404	93.0342	5.251	C_7_H_6_O_3_	isomer of 3,4-dihydroxybenzaldehyde	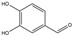	6.44/11.09	nd/nd	[13]
12.	9.97	139.03783	111.0433	−8.204	9.32	137.02316	120.8092,103.4394	−1.172	C_7_H_6_O_3_	isomer of 3,4-dihydroxybenzaldehyde	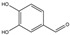	nd/nd	0.48/3.86	[13]
13.	10.45	317.12305	**299.1147**,281.1026,263.0923	−0.139	10.28	315.10757	153.0561,**135.0455**	0.400	C_14_H_20_O_8_	isomer of hydroxytyrosol glucoside	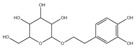	nd/1.50	nd/nd	[14]
14.	nd	nd	nd	nd	10.69	353.08662	191.0551,179.0340	−0.251	C_16_H_18_O_9_	isomer of chlorogenic acid	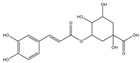	nd/0.34	nd/nd	[15]
15.	nd	nd	nd	nd	11.20	341.08682	179.0339	0.327	C_15_H_18_O_9_	isomer of caffeic acid 3-glucoside	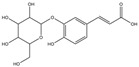	nd/0.90	nd/nd	[14]
16.	nd	nd	nd	nd	11.43	375.12872	**213.0763**,125.0603	0.391	C_16_H_24_O_10_	loganic acid	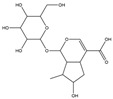	nd/5.36	nd/3.21	[14]
17.	nd	nd	nd	nd	11.62	389.10786	345.1174,183.0654	0.057	C_16_H_22_O_11_	secologanoside	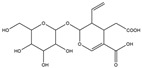	nd/4.81	nd/0.30	[14]
18.	nd	nd	nd	nd	14.68	487.14523	**469.1335**,427.1229,397.1122,233.0445,203.0340,179.0341	1.258	C_21_H_28_O_13_	cistanoside F	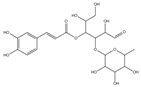	nd/0.70	nd/nd	
19.	14.40	181.04939	163.0387	−0.802	14.80	179.03457	135.0444	3.825	C_9_H_8_O_4_	isomer of caffeic acid	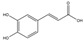	4.05/6.43	nd/0.77	[13]
20.	14.45	463.18079	317.1227,301.1280	−0.459	14.85	461.16580	**315.1071**,205.0709,135.0445	0.970	C_20_H_30_O_12_	forsythoside E	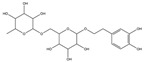	2.62/0.869	nd/0.74	[14]
21.	nd	nd	nd	nd	15.00	353.08679	191.0551	0.231	C_16_H_18_O_9_	isomer of chlorogenic acid	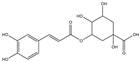	nd/0.92	nd/0.20	[16]
22.	nd	nd	nd	nd	17.73	493.13428	475.1233	0.461	C_23_H_26_O_12_	derhamnosyl suspensaside	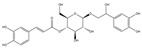	nd/0.43	nd/nd	[14]
23.	nd	nd	nd	nd	17.75	163.03867	146.3310,135.0436,118.9913	−1.844	C_9_H_8_O_3_	isomer of p-coumaric acid	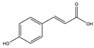	nd/nd	nd/0.16	
24.	18.20	477.13864	459.1276,325.0911,163.0387	−1.044	nd	nd	nd	nd	C_23_H_24_O_11_	plantasioside	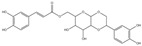	12.66/nd	nd/nd	[14]
25.	18.43	165.05457	147.0435	−0.307	18.67	163.03963	119.0498	4.044	C_9_H_8_O_3_	isomer of p-coumaric acid	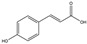	1.91/0.185	nd/0.22	[15]
26.	22.13	625.21222	607.2369,589.2271	−0.762	22.27	623.19714	461.1649,443.1544	−0.150	C_29_H_36_O_15_	forsythoside H	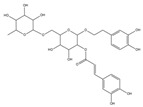	7.48/14.43	1.23/3.15	[14]
27.	nd	nd	nd	nd	23.14	477.13669	315.1053,179.0331,**161.0226**	−5.131	C_23_H_26_O_11_	calceolarioside A	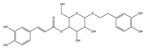	nd/nd	nd/1.03	[14]
28.	23.20	625.21246	479.1540,471.1494,**325.0915**	−0.379	nd	nd	nd	nd	C_29_H_36_O_15_	forsythoside A	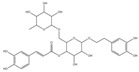	7.43/nd	nd/nd	[14]
29.	23.40	623.19672	**477.1382**,325.0912	−0.524	23.22	621.18140	**487.1441**,469.1337,459.1494	0.005	C_29_H_34_O_15_	suspensaside A	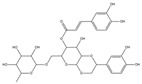	1.08/1.80	nd/nd	[14]
30.	nd	nd	nd	nd	24.49	519.18640	**357.1328**	0.601	C_26_H_32_O_11_	pinoresinol-4′-O-glucopyranoside	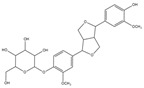	nd/5.39	nd/3.20	[14]
31.	24.80	189.11209	171.1012,125.0957	−0.241	24.67	187.09706	125.0967	3.071	C_9_H_16_O_4_	azelaic acid	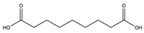	2.32/9.54	nd/4.80	[15]
32.	25.89	521.20148	359.1483	−0.495	26.20	519.18646	357.1331	0.716	C_26_H_32_O_11_	epipinoresinol-4′-β-D-glucopyranoside	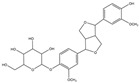	0.37/4.07	0.19/3.96	[14]
33.	26.01	375.14349	357.1327	−0.905	26.12	373.12838	343.1174,313.1071	0.538	C_20_H_22_O_7_	(+)-1-hydroxylpinoresinol	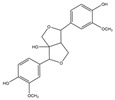	11.87/6.69	4.25/2.13	[14]
34.	26.08	359.14876	**341.1375**,223.0961,137.0594	−0.431	nd	nd	nd	nd	C_20_H_22_O_6_	isomer of pinoresinol	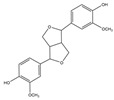	6.16/nd	2.16/nd	[2]
35.	29.23	389.15921	**371.1485**,247.0963,217.0858	−0.693	29.52	387.14408	**372.1202**	0.647	C_21_H_24_O_7_	forsythialan B	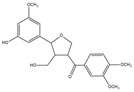	16.66/0.81	7.12/nd	[2]
36.	31.11	335.22156	317.2100,**289.2160**	−0.376	nd	nd	nd	nd	C_20_H_30_O_4_	isomer of dehydropinifolic acid	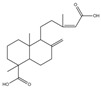	2.54/nd	2.65/nd	[2]
37.	31.49	359.14883	**341.1378**,223.0962,137.0595	−0.236	31.46	357.13330	342.1093,**313.1433**,209.0810	0.098	C_20_H_22_O_6_	matairesinol	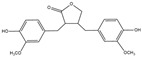	3.44/1.79	0.54/0.59	[14]
38.	32.70	359.14868	341.1376,311.1272,235.0961,217.0858	−0.654	nd	nd	nd	nd	C_20_H_22_O_6_	isomer of pinoresinol	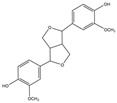	1.07/nd	2.92/nd	[2]
39.	32.78	335.22141	**317.2107**,299.2003	−0.823	nd	nd	nd	nd	C_20_H_30_O_4_	isomer of dehydropinifolic acid	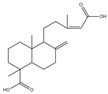	2.22/nd	2.68/nd	[2]
40.	32.81	359.14627	341.1359,**235.0946**	−7.364	nd	nd	nd	nd	C_20_H_22_O_6_	isomer of pinoresinol	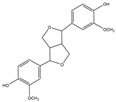	nd/nd	1.19/nd	[2]
41.	33.55	335.22177	**317.2105**,299.2001,289.2157	0.251	nd	nd	nd	nd	C_20_H_30_O_4_	isomer of dehydropinifolic acid	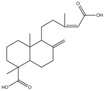	2.39/nd	2.51/nd	[2]
42.	34.45	335.22156	**317.2102**,299.1998,287.1999	−0.376	nd	nd	nd	nd	C_20_H_30_O_4_	isomer of dehydropinifolic acid	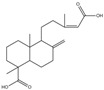	1.05/nd	3.33/nd	[2]

Exptl. [M + H]^+^ represents the experimental *m*/*z* of a molecular ion in positive mode; Exptl. [M − H]^−^ represents the experimental *m*/*z* of a molecular ion in negative mode. Ref: references; WE: water extracts of Lianqiao; EE: ethanol extracts of Lianqiao; nd not detectable.

## Data Availability

The original contributions presented in the study are included in the article/Appendix A, further inquiries can be directed to the corresponding author/s.

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
