# Peer review of "Chemical Compositions of Lianqiao (Forsythia suspensa) Extracts and Their Potential Health Benefits"

_pharmaceuticals, 2024, doi:10.3390/ph17060740_

Round 1
Reviewer 1 Report
Comments and Suggestions for Authors
The manuscript investigates the chemical components of Forsythia suspensa water and ethanol extracts. The study examines their inhibitory effects on SARS-CoV-2 virus spike protein binding to ACE2, inhibition of ACE2 activity, and their capacity to scavenge free radicals. A total of 42 compounds were identified in the extracts using HPLC-MS/MS analysis. The water extract showed greater ACE2 inhibition but weaker inhibition of the SARS-CoV-2 spike protein binding to ACE2 compared to the ethanol extract. The water extract also had a higher phenolic content and greater radical scavenging capacities. The main comments are shown below:
1. Table 1, compounds ID number 5, didn’t include references.
2. Table 1 shows many peaks with the same compound names and structures but with different IDs (rotation time), e.g. 11 and 12How did that happen?
3. Table 1, some compounds didn’t give the rotation time.
4. There is a famous component from Forsythia suspensa, forsythin, but it didn’t show in your results. I’m wondering if there are any issues with the extraction or detection.
5. The author should add the LC-MS traces of the two extractions.
6. Line 210, all the compound numbers should be bolded.
Author Response
1. Reviewer #2:Table 1, compounds ID number 5, didn’t include references.
Response: Thank you for the comment. As the reviewer may have observed, in addition to compound #5, several other compounds (#18, #23) also didn’t include references. Since these compounds were detected and identified in Lianqiao for the first time, and no references could be found to support their existence. Therefore, the identification of these compounds relied on chemical databases such as SciFinder and ChemSpider. To address this, a sentence has been added to the compound identification section to clarify this point (Line 176-179 in Material and Methods Part): The chemical compounds in Lianqiao water and ethanol extracts were determined by their molecular weights, mass fragment information, previously published related manuscripts, and chemical databases including SciFinder and ChemSpider.
2. Reviewer #2:Table 1 shows many peaks with the same compound names and structures but with different IDs (rotation time), e.g. 11 and 12. How did that happen?
Response: Due to their identical elemental composition, molecular formula, and analogous fragment information, it is challenging to attribute specific substances, leading them to be labeled as "Isomer" in this article. Consequently, substances with the same molecular weight but different retention time may all be denoted as isomers of a compound. Meanwhile, a sentence has been included in the structural identification section of the manuscript to address this issue: Some identified compounds had the same elemental composition, molecular formula, and similar fragment information, making it impossible to determine their specific substance attribution, and they were all marked as Isomer in this article. (Line 91-94 in the Results and Discussion Part).
3. Reviewer #2:Table 1, some compounds didn’t give the rotation time.
Response: No change was made. Some compounds can only be detected in positive ion mode, and as a result, they do not have a retention time in negative ion mode, and vice versa.
4. Reviewer #2: There is a famous component from Forsythia suspensa, forsythin, but it didn’t show in your results. I’m wondering if there are any issues with the extraction or detection.
Response: No change was made. It was noted that forsythin was not detected from the Lianqiao extracts in our study. Upon reviewing previous studies, it was found that forsythin was predominantly extracted from a methanol-water mixture, which differs from the pure water heating or ethanol extraction methods employed in this study. This difference may account for the inability to detect the substance. We are confident that our extraction method and compound identification process have effectively addressed major substances in this extract, including other representative compounds in Lianqiao, such as forsythoside.
5. Reviewer #2:The author should add the LC-MS traces of the two extractions.
Response: I am unsure whether the reviewer is referring to the typical LC-MS chromatograms of the samples. Representative LC-MS chromatograms of Lianqiao water and ethanol extracts, obtained in both ESI positive and negative ion modes, have been included in the supplementary materials (Figure S1).
6. Reviewer #2:Line 210, all the compound numbers should be bolded.
Response: Done (Line 100-101 in the Results and Discussion Part).

Reviewer 2 Report
Comments and Suggestions for Authors
The article concerns the study of the fruits of the Lianqiao (Forsythia suspensa). The authors investigated the chemical constituents of its water and ethanol extracts using HPLC-MS/MS analysis. They have studied: 1) inhibition of SARS-CoV-2 virus spike protein binding to ACE2 by the extracts; 2) inhibition of ACE2 activity as well as the capacity to scavenge free radicals. The authors have identified 42 metabolites, preferably phenolic compounds were tentatively identified in the extracts by. The water extract revealed a greater ACE2 inhibition, but a weaker inhibition on SARS CoV-2 spike protein binding to ACE2. They also have determined the phenolic content and found that the water extract contained more phenolic compounds than ethanolic one. They found that the water extract is greater free radical scavenger than ethanolic extract. The authors concluded about high potential COVID-19 preventive applications of the Lianqiao and its extracts.
The article seems to be interesting for the journal and the methodology seems to be adequate. Several minor imperfections:
1. Line 20. Replace “Forsythia suspensa (Lianqiao)” with “…the Lianqiao (Forsythia suspensa)” in order to avoid the taxonomical confuse – Lianquao as an author of the species and write please the Latine name in Italic.
2. Line 195. Write, please, Latin name by Italic.
3. Nable 1. The title formatting should be fixed, the formulae should be enlarged or shifted on separate figures. The names of the substances should be written from the first small letters.
4. Fig. 1. The formulae should be enlarged.
5. Line 304. “13.2 - 13.9 mg” should be replaced with “13.2–13.9 mg”
6. The main flaw of the article is a disruption between the discussion on biological activities and chemical analysis of two different extracts. The authors didn’t attempt to explain the differences in the activities by the differences in the content of phenolic compounds although it seems to be obvious for anti-oxidant activity. The discussion should be enforced by this way.
My general opinion: the manuscript may be published after minor revision.
Author Response
1. Reviewer #1:Line 20. Replace “Forsythia suspensa (Lianqiao)” with “…the Lianqiao (Forsythia suspensa)” in order to avoid the taxonomical confuse – Lianquao as an author of the species and write please the Latine name in Italic.
Response: Done (Line 20 in the Abstract Part).
2. Reviewer #1:Line 195. Write, please, Latin name by Italic.
Response: Done. Forsythia suspensa has been written by italic throughout the manuscript.
3. Reviewer #1:Table 1. The title formatting should be fixed, the formulae should be enlarged or shifted on separate figures. The names of the substances should be written from the first small letters.
Response: The chemical structures presented in this study adhere to the ACS document 1996 format, with a resolution exceeding 300 dpi. We enlarge the final PDF file to ensure the readability of these chemical structures. It is deemed unnecessary to individually allocate one page for each chemical structure. Moreover, the names of the substances has been written from the first small letters (Table 1).
4. Reviewer #1: 1. The formulae should be enlarged.
Response: No change was made. The formulae is large enough to be clearly seen, and it is unnecessary to be enlarged.
5. Reviewer #1:Line 304. “13.2 - 13.9 mg” should be replaced with “13.2–13.9 mg”
Response: Done (Line 95 in the Results and Discussion Part).
6. Reviewer #1:The main flaw of the article is a disruption between the discussion on biological activities and chemical analysis of two different extracts. The authors didn’t attempt to explain the differences in the activities by the differences in the content of phenolic compounds although it seems to be obvious for anti-oxidant activity. The discussion should be enforced by this way.
Response: Done. The related discussion has been added in the revised manuscript (Line 123-124 in the Results and Discussion Part)

Round 2
Reviewer 1 Report
Comments and Suggestions for Authors
The authors successfully revised the manuscript, which is now suitable for publication.